# The mechanism of aerobic exercise in ameliorating glycolipid metabolic disorders in metabolic syndrome rats via the miR-27a-PPARγ pathway

Peiyun Wu[1☯], Zhizhuo Wang[1☯], Kunhui Li[2], Yao Gao[3], Jing Yang[1], Jinwu Wang[4], Chunlei Shan[5], Qi Guo[6], Han Yang[7], Juan Wang[1], Yuanshan Jiang[1], Cheng Lin[1]*

1 Department of Rehabilitation Medicine, School of Health, Fujian Medical University, Fuzhou, China, 2 Department of Rehabilitation, The First Affiliated Hospital of Fujian Medical University, Fuzhou, China, 3 Department of Medical Laboratory, School of Medical Technology and Engineering, Fujian Medical University, Fuzhou, China, 4 Department of Orthopedics, School of Medicine, Shanghai Key Laboratory of Orthopedic Implant, Ninth People's Hospital Affiliated to Shanghai Jiao Tong University, Shanghai, China, 5 Department of Rehabilitation, School of Medicine, Shanghai Jiao Tong University, Shanghai, China, 6 Department of Rehabilitation Sciences, School of Medicine and Health Sciences, Shanghai University, Shanghai, China, 7 Department of Medical Laboratory, School of Biomedical Engineering, Shanghai Jiao Tong University, Shanghai, China

☯ These authors are contributed equally to this work.
* fjmulc@fjmu.edu.cn

## Abstract

This study investigated how aerobic exercise improves metabolic disorders in rats with metabolic syndrome (MS) and explored the role of miR-27a in this process. MS was induced by a high-fat and high-sugar diet. After eight weeks of treadmill training, aerobic exercise was found to reduce metabolic abnormalities and decrease elevated miR-27a levels, while increasing the expression of PPARγ and key downstream molecules involved in glycolipid metabolism. Downregulating miR-27a via a viral vector produced benefits similar to those of exercise, and combining miR-27a inhibition with exercise led to further improvement. These results suggested that aerobic exercise alleviates MS-related metabolic disorders partly through suppressing miR-27a and promoting PPARγ signaling, revealing a potential therapeutic target.

## Introduction

A collection of metabolic disorders known as metabolic syndrome (MS) are characterized by insulin resistance, aberrant lipid metabolism, abdominal obesity, hypertension, and reduced glucose tolerance [1]. Over time, there has been a significant increase in the prevalence of MS. Evidence suggests that MS has a major detrimental impact on people's health and quality of life, and it heavily strains both public and private health systems financially [2]. The etiology of MS is intricate and unclear. Numerous factors, including genetics, environment, immunity, and an irregular

**Data availability statement:** All relevant data are within the paper.

**Funding:** This study was funded by The Education and Research Project of Young and Middle-aged Teachers in Fujian Province (Science and Technology) (JAT220095 to ZW) and The Natural Science Foundation of Fujian Province of China (2020J01654 to CL).

**Competing interests:** The authors have declared that no competing interests exist.

lifestyle, might contribute to MS [1]. Multiple studies have demonstrated the positive effects of aerobic exercise on improving MS [3–5]. Since it increases lipolysis and fatty acid oxidation while decreasing the hazardous buildup of lipids in liver and skeletal muscle, aerobic exercise is important for the metabolism of glycolipids [6]. Still, it is not known exactly what molecular pathways are at play.

MicroRNAs (miRNAs) are short, single-stranded RNAs that are 19–23 nucleotides in length. They regulate gene expression and are created by transcription of DNA. A wide range of disorders can occur and worsen as a result of abnormal expression of miRNAs [7]. Numerous diseases, such as those involving the metabolism of glycolipids, cardiovascular conditions, and disorders of the bones, are intimately linked to the expression of miR-27a [8]. The family of nuclear receptors and transcription factors known as peroxisome proliferator-activated receptors (PPARs) are ligand-activated. PPARs are classified into three isoforms: PPAR α, β/δ, and γ, of which PPARγ is crucial for energy homeostasis, lipid metabolism, and insulin sensitivity [9,10]. Recent studies have also revealed a substantial correlation between miR-27a and PPARγ. Exercise-induced browning of white adipose tissue and improved insulin sensitivity in skeletal muscle is facilitated by exosomal miR-27a, which activates the PPARγ-stimulated insulin receptor substrate-1 (IRS-1)/Akt/Glucose transporter type 4 (GLUT4) signaling pathway [11]. It was proven that downregulated miR-27a expression can affect its target gene PPARγ, which regulates adipocyte differentiation [12,13]. By blocking PPARγ, Yu et al. demonstrated that adipose tissue-derived miR-27a may be crucial for the emergence of obesity-induced insulin resistance in skeletal muscle [14]. In contrast, excessive expression of miR-27a particularly suppresses adipocyte growth by preventing the production of PPARγ [15].

In brief, glycolipid metabolism disorders are significant components of MS. Notably, PPARγ is critical for maintaining glucose and lipid metabolism homeostasis, and overexpression of miR-27a leads to blockage of PPARγ expression. Furthermore, aerobic exercise markedly improves metabolic disorders in MS patients. Therefore, we proposed that aerobic exercise improves glycolipid metabolism by suppressing miR-27a overexpression, increasing PPARγ expression, and regulating downstream insulin signaling-related GLUT4 and IRS-1 expression in skeletal muscle and downstream lipid processing-involved enzymes such as carnitine palmitoyltransferase 1 (CPT1) and long-chain acyl CoA dehydrogenase (LCAD) expression in liver, which subsequently improved metabolic disorders.

## Methods

### Animals, diets and interventions

Six-week-old male Sprague–Dawley (SD) rats weighing 160−200 g were used. The reporting of this study complies with ARRIVE recommendations. This research received approval from Fujian Medical University's Animal Ethics Review Committee (No. FJMU IACUC 2021−0329) and was conducted following the guidelines outlined in the National Institutes of Health Guide for the Care and Use of Laboratory Animals.

Following the guidelines for laboratory animals, rats were housed under conditions of 25±2°C temperature, 55±5% humidity, and a 12-hour light/12-hour dark

cycle, with free access to food and water. The rats in the control groups were fed a standard chow diet. A diet heavy in fat and sugar was used to induce the MS model. The MS model was determined when two of the following criteria were satisfied: 1) the body weight in the MS group was greater than the mean body weight plus twice the standard deviation in the control group; 2) the fasting blood glucose ≥5.6 mmol/L or random blood glucose ≥6.1 mmol/L; 3) the blood pressure was > 130/85 mmHg; 4) the triglyceride level was > 1.7 mmol/L, and the high-density lipoprotein cholesterol (HDL-C) level was < 1.0 mmol/L [16].

Rats in the down-regulated miR-27a group and the null group received tail vein injections of adeno-associated virus containing the miR-27a sponge and null adeno-associated virus, respectively. The exercise group followed an 8-week treadmill exercise regimen based on an incremental intensity training program from a previously documented protocol [17], whereas the quiet group used the treadmill at rest for the same period (The grouping of the two experiments shown in Fig 1A-B). During treadmill training, if a rat exhibited strong resistance (freezing, rearing, or vocalization), the belt was stopped immediately and the animal was returned to its home cage for at least 30 min before a second attempt; after three consecutive failures the day's session was cancelled.

The sample size in this study was determined based on the following principles: (1) To account for individual variability and the risk of failure in establishing the high-fat and high-sugar diet-induced model, more animals were initially included in the modeling group to ensure that a sufficient number of successful model animals would be available for subsequent grouping; (2) In accordance with the ARRIVE 2.0 guidelines [18], when pilot data are unavailable, referencing existing literature is a valid approach for determining sample size. Studies in the field of metabolism commonly employ sample sizes of 5–8 animals per group, which has been proven effective in detecting statistically significant differences in key metabolic parameters [19,20]; (3) Strict adherence to the 3R principles was maintained by controlling for factors such as animal strain, sex, age, and rearing conditions to minimize within-group variability and reduce animal usage.

## Basic indicators measurement

Before the experiment began, the team ensured proficiency in basic indicator measurements so that each test could be completed as quickly as possible to minimize distress to the rats. The body weight, length, fasting blood glucose, and systolic blood pressure in the tail artery of the rats were measured every two weeks. The formula for Lee's index is: (body weight × 1000)^(1/3)/body length (cm) [21]. The fasting blood glucose was measured in rats after fasting for more than 8 h using a rapid glucose tester. The systolic blood pressure of conscious rats was evaluated using a noninvasive tail-cuff technique.

## Sample collection and processing

The rats were fasted for at least 12 hours prior to euthanasia. For euthanasia, animals were deeply anesthetized via an intraperitoneal injection of sodium pentobarbital at a dose of 0.2g per 100g body weight. The depth of anesthesia was confirmed by the absence of pedal and corneal reflexes. Following induction of anesthesia, blood samples were collected from the abdominal aorta. Subsequently, euthanasia was completed by cervical dislocation. No additional analgesics were administered, as the animals were under deep anesthesia prior to tissue collection and did not recover consciousness. Fasting glucose and serum lipids, including triglyceride, HDL-C, and low-density lipoprotein cholesterol (LDL-C), were measured with an automated biochemical analyzer. The serum insulin concentration was determined via an enzyme-linked immunosorbent assay. The homeostatic model assessment for insulin resistance (HOMA-IR) was calculated by the formula: HOMA-IR = fasting insulin (μU/dL) × fasting glucose (mmol/L)/22.5 [22]. The gastrocnemius muscle and liver were then removed and stored at −80 °C.

## Real-time polymerase chain reaction (PCR) for detecting mRNA expression

Total RNA was extracted from tissues using Freezol reagent. The reverse transcription reaction system was prepared according to the Hifair® III 1st Strand cDNA Synthesis SuperMix for qPCR kit instructions. The relative expression of each target gene was calculated using the $2^{-\Delta\Delta Ct}$ method. The detailed primer sequences are shown in Table 1.

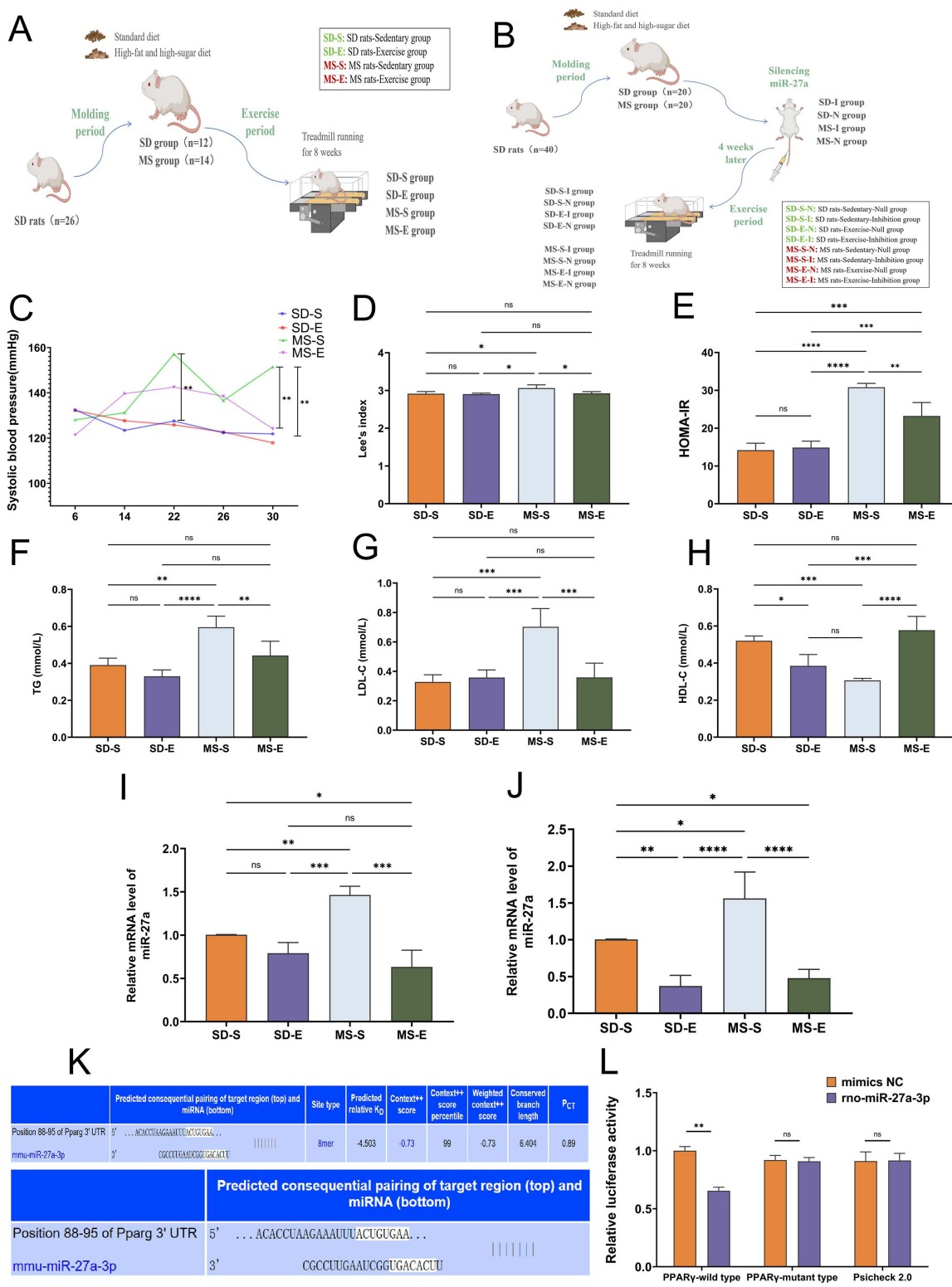

**Fig 1. Grouping of the two experiments, systolic blood pressure variations, Lee's index, HOMA-IR, serum lipids levels in rats in each group, mRNA expression level in liver and skeletal muscle of rats in each group, gene prediction by Target Scan database and results of the miR-27a-targeted PPARγ dual luciferase assay (MS-E, n=5; MS-S, n=7; SD-E, n=5; SD-S, n=5). (A-B)** Experiment 1 study design and experiment 2

study design. **(C)** The levels of systolic blood pressure in each group of rats from the beginning to the end of the experiment. **(D)** Lee's index of rats in each group at the end of the experiment. **(E)** HOMA-IR levels of rats in each group at the end of the experiment. **(F-H)** Serum lipids content of rats in each group. (I) miR-27a mRNA expression levels in liver of rats. (J) miR-27a mRNA expression levels in the gastrocnemius muscle of rats. **(K)** Gene prediction results by using the Target Scan database. **(L)** Results of the miR-27a-targeted PPARγ dual luciferase assay. Statistical analyses were performed using the one-way ANOVA followed by LSD post-hoc to compare variances. Data represented mean ± standard deviation. $^*P<0.05$; $^{**}P<0.01$; $^{***}P<0.001$; $^{****}P<0.0001$; ns, not significant.

## Western blotting for detecting protein expression

Protein lysates were prepared from tissues. The target proteins were separated using 10% sodium dodecyl sulfate polyacrylamide gel electrophoresis and then transferred to nitrocellulose membranes. The blotted membranes were incubated with primary antibodies overnight at 4°C, followed by incubation with horseradish peroxidase-coupled secondary antibodies for 1 h at room temperature. Signal analysis of the protein blots was performed with a chemiluminescence instrument. The blots were cut before hybridization with antibodies due to budgetary restrictions and an attempt to preserve antibodies.

## Clarifying whether miR-27a directly targets PPARγ by gene prediction

Based on the PCR-based Accurate Synthesis method, we designed full-length overlapping primers to synthesize the PPARγ-wild type and PPARγ-mutant type sequences through two rounds of PCR. These sequences were then cloned into the psicheck2.0 vector, resulting in the recombinant vectors PPARγ-wild type and PPARγ-mutant type. Concurrently, rno-miR-27a-3p mimics were synthesized in vitro. The co-transfection into 293T cells was performed using the Lipo3000 lipid transfection method, and the regulatory effects of rno-miR-27a-3p mimics on the PPARγ gene were evaluated by detecting the luminescence values of firefly luciferase and kidney luciferase.

## Immunohistochemical analysis of skeletal muscle GLUT4

Skeletal muscle stored in tissue fixative was dehydrated, embedded, and cut into wax blocks. The slices were dewaxed, repaired and closed, and GLUT4 antibody, secondary antibody and color developer were added. The tissue was observed under a microscope and scanned. Then, the positive areas in each group were quantitatively analyzed.

**Table 1. Primer sequences for RT–qPCR.**

| Target gene | Primer Sequences |
| --- | --- |
| *MiR-27a* | Upstream primer: 5'-TTCCGCGTTCACAGTGGCTAAG-3'<br>Downstream primer: 5'-AGTGCAGGGTCCGAGGTATT-3' |
| *U6* | Upstream primer: 5'-CTCGCTTCGGCAGCACATATACT-3'<br>Downstream primer: 5'-ACGCTTCACGAATTTGCGTGTC-3' |
| *PPARγ* | Upstream primer: 5'-AGCCCTTTGGTGACTTTATGG-3'<br>Downstream primer: 5'-CAGCAGGTTGTCTTGGATGT-3' |
| *GLUT4* | Upstream primer: 5'-CCAGTATGTTGCGGATGCTATG-3'<br>Downstream primer: 5'-TGGTTTCAGGCACTCTTAGGA-3' |
| *IRS-1* | Upstream primer: 5'-GTGGAGTTGAGTTGGGCAGA-3'<br>Downstream primer: 5'-CCTGTCCGCATGTCAGCATA-3' |
| *LCAD* | Upstream primer: 5'-GTTCGATTGCCAGCTAGTGC-3'<br>Downstream primer: 5'-ACAGTCTGGATGTGTGCGAC-3' |
| *CPT1* | Upstream primer: 5'-TTCCGGTTCAAGAATGGCATC-3'<br>Downstream primer: 5'-TCGTCTGGCTTGACATTCGG-3' |
| *β-actin* | Upstream primer: 5'-CGCGAGTACAACCTTCTTGC-3'<br>Downstream primer: 5'-CCTTCTGACCCATACCCACC-3' |

### HE staining of liver and skeletal muscle tissue

The liver and skeletal muscle tissues stored in tissue fixative were dehydrated and dipped in wax, and then the dipped tissues were embedded in an embedding machine, and the wax blocks were trimmed for later use. The wax blocks were sliced with a paraffin slicer, then dewaxed, repaired and sealed, stained with hematoxylin and eosin, dehydrated and sealed, observed under a microscope, scanned, and the images were captured for analysis.

### Statistical analysis

The data were analyzed using SPSS 26.0 and are presented as the mean ± standard deviation. For differences between two groups, analyses were performed using the Student's t-test; between three or more groups, analyses were performed using one- or two-way analysis of variance followed by a post hoc least-significant difference (LSD) test to determine individual differences between groups. $P$ values <0.05 were considered to indicate statistical significance.

## Results

### Evaluation of MS rats model constructed by feeding high-fat and high-sugar diets

As shown in Table 2, in experimentation 1, tail artery systolic blood pressure, diastolic blood pressure, fasting blood glucose, body weight and Lee's index were significantly higher in MS rats than in SD rats ($P < 0.001$, $P < 0.01$, $P < 0.05$, $P < 0.05$, $P < 0.05$). In this case, the criteria for blood pressure and fasting blood glucose were in accordance with the MS modeling criteria. A total of 22 rats (MS-E, n=5; MS-S, n=7; SD-E, n=5; SD-S, n=5) that fulfilled the modeling criteria and survived the treadmill protocol were included in the final analysis.

### Aerobic exercise improves Lee's index, HOMA-IR, systolic blood pressure and serum lipids in MS rats

As shown in Fig 1, the baseline systolic blood pressure was measured, and there was no significant difference among the four groups. Until the end of the experiment, the systolic blood pressure in the MS-S group was always greater than that in the SD-S group. After 8 weeks of exercise, the systolic blood pressure in the MS-E group was considerably lower compared with the MS-S group. Nonetheless, the systolic blood pressure of the SD-S rats did not markedly change throughout the trial. The systolic blood pressure of SD-E rats decreased after aerobic exercise training. Therefore, aerobic exercise significantly reduced hypertension in the MS-E group and had a significant hypotensive effect on SD-E rats (Panel C).

Prior to the trial commencement, the body weight and length of each group were measured as a benchmark, and no discernible variation was found. As shown in Panel D, the Lee's index of the MS-S group was greater than that of the SD-S group ($P < 0.05$). In addition, the Lee's index was significantly lower in the MS-E group than in the MS-S group ($P < 0.05$). Therefore, a high-fat and high-sugar diet may increase the Lee's index, and aerobic exercise may attenuate the increase in the Lee's index.

**Table 2. MS rat modeling validation.**

| Group | SD group (n=10) | MS group (n=12) |
|---|---|---|
| Systolic blood pressure (mmHg) | 125.70 ± 13.74 | 151.06 ± 16.66*** |
| Diastolic blood pressure (mmHg) | 89.37 ± 15.14 | 107.44 ± 12.87** |
| Fasting blood glucose | 5.13 ± 0.46 | 5.65 ± 0.54* |
| Body weight (g) | 511.60 ± 61.20 | 568.64 ± 58.80* |
| Lee's index | 2.79 ± 0.12 | 2.90 ± 0.11* |

Statistical analyses were performed using the Student's t-test to compare variances. Data represented mean ± standard deviation. *$P < 0.05$, **$P < 0.01$, ***$P < 0.001$ compared to SD group.

The results revealed that HOMA-IR was significantly greater in the MS-S group ($30.82 \pm 0.88$) than in the SD-S group ($14.19 \pm 1.58$). Moreover, the HOMA-IR score in the MS-E group ($23.23 \pm 3.08$) was significantly greater than that in the SD-E group ($14.87 \pm 1.46$) ($P < 0.001$). For MS rats, HOMA-IR was significantly greater in the MS-E group than in the MS-S group ($P < 0.01$) (Panel E).

Triglyceride and LDL-C were significantly greater, and HDL-C was significantly lower in the MS-S group than in the SD-S group. Nevertheless, aerobic exercise significantly increased HDL-C and decreased triglyceride and LDL-C in the MS-E group compared with the MS-S group (Panel F-H).

## Aerobic exercise reduced mRNA expression of miR-27a in liver and skeletal muscle of MS rats

As shown in Fig 1, MiR-27a mRNA expression in the MS-S group appeared to be greater than that in the SD-S group ($P < 0.01$), suggesting that elevated miR-27a expression in liver of rats might be accountable for the development of MS. In MS rats, the expression of miR-27a mRNA was significantly greater in the MS-S group than that in the MS-E group, suggesting that 8 weeks of aerobic exercise suppressed the expression of miR-27a mRNA in skeletal muscle in the MS-E group (Panel 1I). Similarly, the expression of miR-27a in skeletal muscle was similar to that in liver: miR-27a mRNA expression was greater in the MS-S group than that in the SD-S group ($P < 0.05$). Compared with that in the MS-S group, miR-27a mRNA expression was lower in livers of rats in the MS-E group (Panel J).

## MiR-27a inhibited PPARγ expression

As shown in Fig 1, miR-27a was predicted to be able to bind to the positional sequence at bases 88−95 of the coding region of the rat PPARγ gene sequence (ACUGUGAA) via the Target Scan database (https://www.targetscan.org/mmu_80/) (Panel K). Further validation of whether miR-27a targets PPARγ by miR-27a-targeted PPARγ dual luciferase assay. PPARγ-wild type-3'Luc was cotransfected with mmu-miR-27a, which had a significantly lower fluorescence intensity than was observed for PPARγ-wild type-3'Luc cotransfected with NC ($P < 0.01$). However, for the PPARγ mutated with the miR-27a complementary sequence, there was no significant difference in fluorescence intensity in the PPARγ-mutant-3'Luc and mmu-miR-27a groups compared to the PPARγ wild-type-3'Luc and mmu-miR-27a groups (Panel L).

## The skeletal muscle of MS rats exhibited elevated PPARγ, GLUT4 and IRS-1 mRNA and protein expression after aerobic exercise

As shown in Fig 2, the expression level of PPARγ mRNA was significantly lower in the MS-S group than that in the SD-S group ($P < 0.05$), and although PPARγ protein expression was not statistically different, it also showed an increasing trend, indicating that the high-fat and high-sugar diet contributed to the reduced expression of PPARγ in skeletal muscle of rats. Specifically, the mRNA and protein expression levels of PPARγ in the MS-E group were greater than those in the MS-S group ($P < 0.05$, $P < 0.001$). In SD rats, the expression level of PPARγ mRNA was also significantly greater in the SD-E group than in the SD-S group ($P < 0.001$) (Panel A-C).

Similarly, a long-term high-fat and high-sugar diet induced the downregulation of GLUT4 and IRS-1 expression in skeletal muscle of rats, as demonstrated in panel D-I. The expression levels of GLUT4 mRNA and IRS-1 protein in the MS-S group were lower than those in the SD-S group ($P < 0.01$, $P < 0.05$). The lack of statistically significant differences in GLUT4 protein expression and IRS-1 mRNA levels between the MS-S and SD-S groups may be attributed to the small sample size. The MS-E and MS-S groups exhibited statistically significant differences in GLUT4 protein ($P < 0.05$), IRS-1 mRNA ($P < 0.01$) and IRS-1 protein ($P < 0.001$) expression, indicating that aerobic exercise training enhanced GLUT4 and IRS-1 expression in skeletal muscle. Aerobic exercise training effectively increased GLUT4 and IRS-1 expression in skeletal muscle, as evidenced by the significantly greater levels of GLUT4 mRNA, IRS-1 mRNA and IRS-1 protein in the SD-E group than that in the SD-S group.

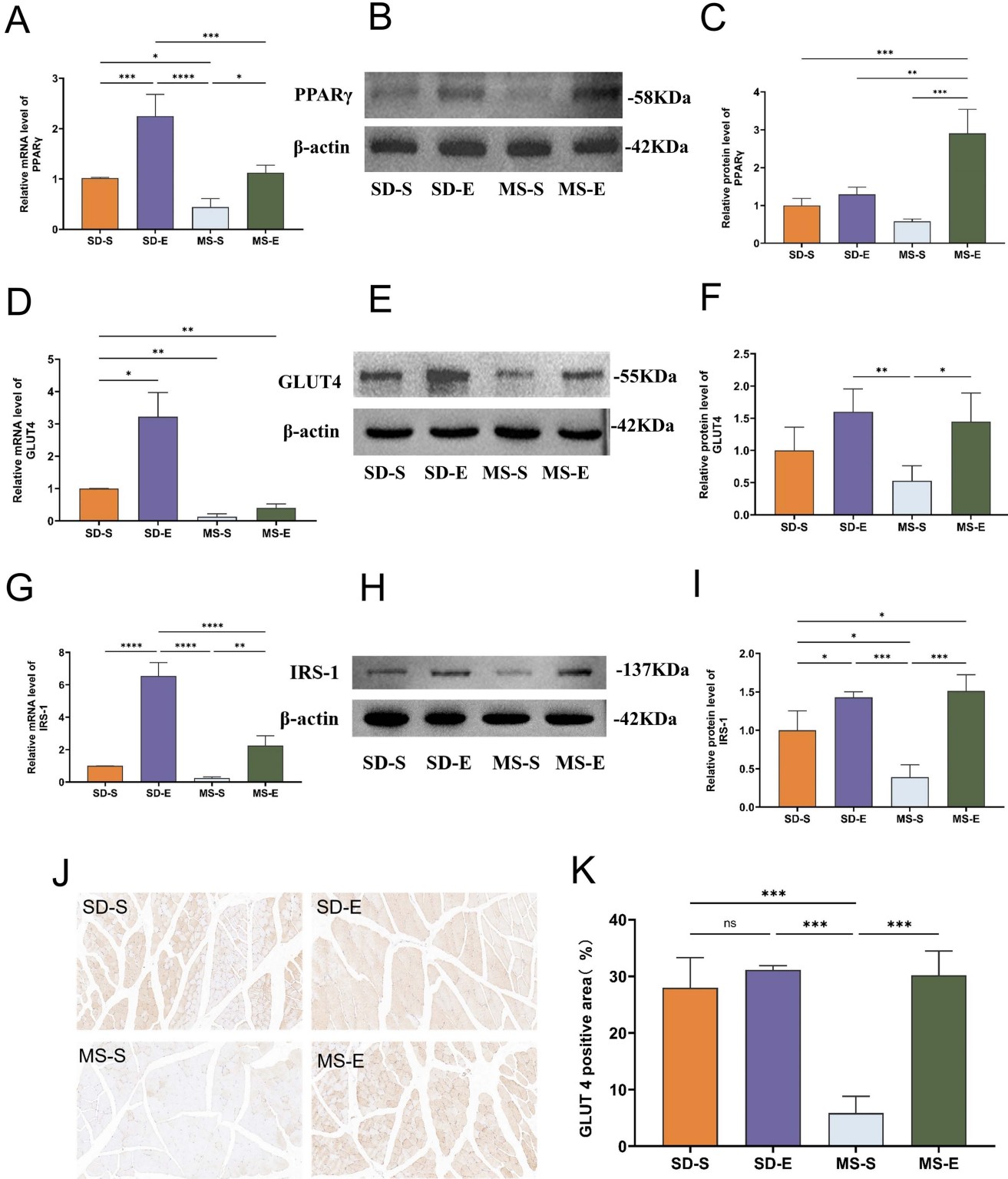

**Fig 2. PPARγ, GLUT4 and IRS-1 mRNA and protein expression level and immunohistochemical quantification of GLUT4 in skeletal muscle (MS-E, n = 5; MS-S, n = 7; SD-E, n = 5; SD-S, n = 5).** **(A)** PPARγ mRNA expression level in gastrocnemius muscle of rats in each group. **(B)** Relative protein expression of PPARγ in each group. **(C)** Immunoblot quantification of PPARγ in each group. **(D)** GLUT4 mRNA expression level in the

gastrocnemius muscle of rats in each group. **(E)** Relative protein expression of GLUT4 in gastrocnemius muscle of rats in each group. **(F)** Immunoblot quantification of GLUT4 in each group. **(G)** Expression level of IRS-1 mRNA in each group. **(H)** Relative protein expression of IRS-1 in each group. **(I)** Immunoblot quantification of IRS-1 in each group. **(J-K)** Skeletal muscle sections of GLUT4-positive rats in each group were analyzed by immunohistochemistry, and the results were shown in brown–yellow. Statistical analyses were performed using the one-way ANOVA followed by LSD post-hoc to compare variances. Data represented mean ± standard deviation. $^{*}P < 0.05$; $^{**}P < 0.01$; $^{***}P < 0.001$; $^{****}P < 0.0001$; ns, not significant.

Increased expression of GLUT4 in skeletal muscle is essential to enhance skeletal muscle glucose uptake. Therefore, in this study, the expression of GLUT4 in skeletal muscle was determined by immunohistochemistry. Skeletal muscle sections with positive GLUT4 expression show a tan color. As shown in panel J-K, the results of immunohistochemical analysis showed that the percentage of GLUT4-positive rats in the MS-S group was lower than that in the SD-S group ($P < 0.001$). On the other hand, the MS-E group exhibited a much greater percentage of positive GLUT4 expression than the MS-S group ($P < 0.001$).

### In liver of MS rats, aerobic exercise increased the expression levels of PPARγ and lipid metabolism-related enzymes mRNA and protein

As seen in Fig 3, compared with those in the SD-S group, the expression level of PPARγ mRNA was significantly lower in the MS-S group ($P < 0.05$). These findings indicated that a high-fat and high-sugar diet lead to a decrease in the expression of PPARγ in liver of rats. The MS-E group had higher levels of PPARγ mRNA and protein expression than that in the MS-S group ($P < 0.01$, $P < 0.01$), indicating that aerobic exercise boosted PPARγ expression in liver of MS rats (Panel A-C).

As shown in panel D-I, compared with those in the SD-S group, the LCAD protein ($P < 0.01$) and CPT1 protein ($P < 0.001$) expression levels were significantly lower. And LCAD mRNA and CPT1 mRNA in the MS-S-N showed a tendency to decrease. Due to insufficient sample size, this result may lack sufficient statistical power to detect differences with small to moderate effect sizes. Moreover, compared with those in the MS-S group, the LCAD, CPT1 mRNA and protein expression levels in the MS-E group were significantly greater ($P < 0.001$). This suggested that aerobic exercise training promoted LCAD and CPT1 mRNA and protein expression in liver of MS rats ($P < 0.001$, $P < 0.01$, $P < 0.001$, $P < 0.01$, $P < 0.0001$, $P < 0.0001$). Compared with those in the SD-S group, the LCAD mRNA ($P < 0.05$) expression levels in the SD-E group was significantly greater.

### Evaluation of MS rats model constructed by feeding high-fat and high-sugar diets and miR-27a silencing in MS rats model

In experimentation 2, after being fed high-fat and high-sugar diets, rats in the high-fat and high-sugar group with tail artery systolic blood pressure greater than or equal to 130 mmHg, tail artery diastolic blood pressure greater than or equal to 85 mmHg, and fasting blood glucose greater than or equal to 5.6 at 32 weeks of age met the criteria for MS (Table 3).

During the experiment, several animals were excluded owing to failure in meeting the modeling criteria or accidental death during treadmill testing. Consequently, data from 36 rats were included in the final analyses: MS-E-I (n = 4), MS-E-N (n = 4), MS-S-I (n = 5), MS-S-N (n = 5), SD-E-I (n = 4), SD-E-N (n = 4), SD-S-I (n = 5), and SD-S-N (n = 5).

In experimentation 2, the skeletal muscle and liver of rats in each group were taken to detect miR-27a mRNA expression. The silencing group could be established as a high-fat and high-sugar diet combined miR-27a silencing model at $P < 0.05$ compared with the null group. As can be seen from Fig 4, miR-27a expression in liver (Panel A) and skeletal muscle (Panel B) of rats in SD-S-I and MS-S-I groups was lower than that in SD-S-N and MS-S-N groups, respectively, and the difference was statistically significant, indicating that miR-27a expression could be effectively suppressed by tail vein injection of adeno-associated virus carrying miR-27a sponge.

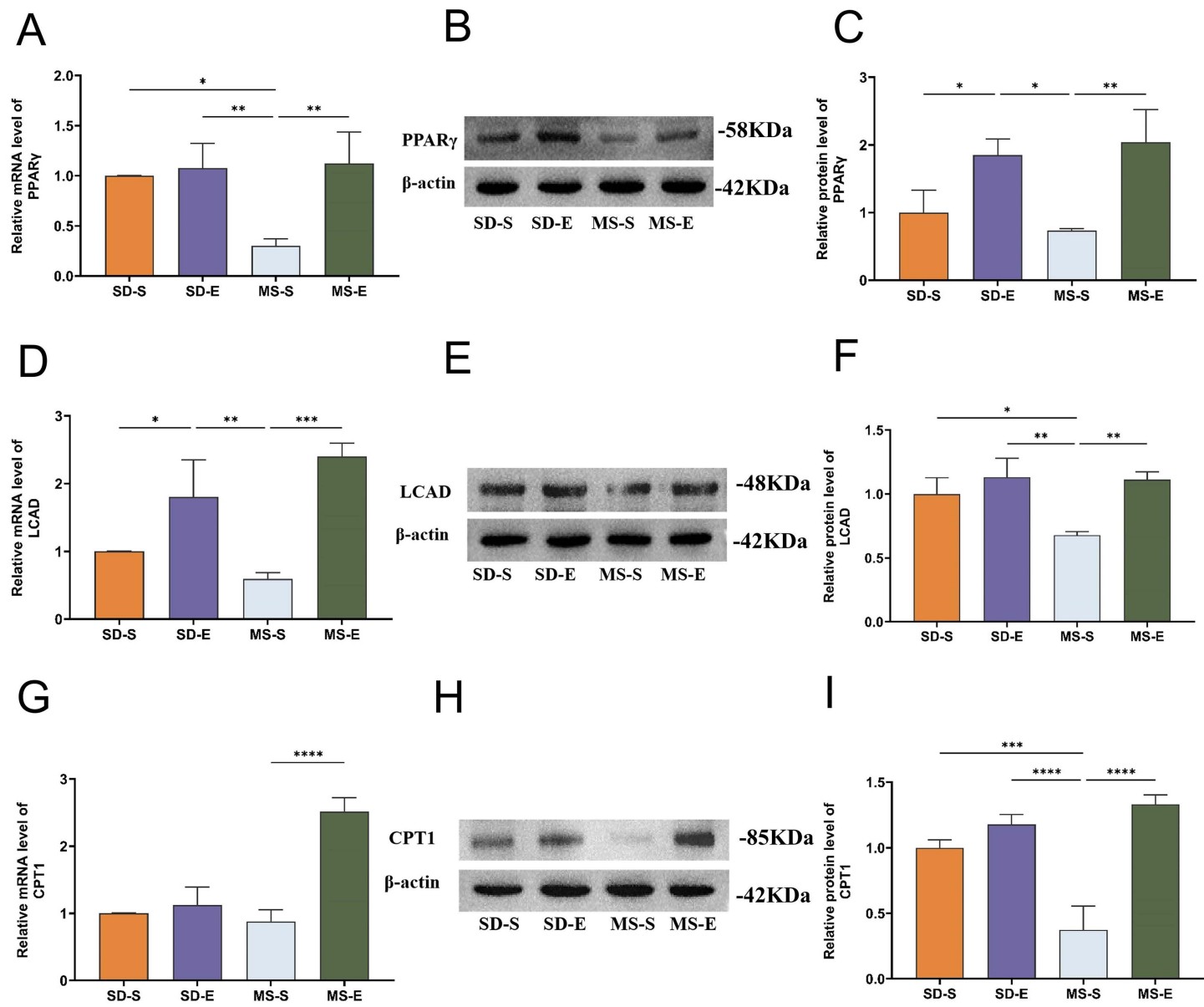

**Fig 3. PPARγ, LCAD and CPT1 mRNA and protein expression levels in liver (MS-E, n=5; MS-S, n=7; SD-E, n=5; SD-S, n=5). (A)** PPARγ mRNA expression level in gastrocnemius muscle of rats in each group. **(B)** Relative protein expression of PPARγ in each group. **(C)** Immunoblot quantification of PPARγ in each group. **(D)** Expression levels of LCAD mRNA in each group. **(E)** Relative protein expression of LCAD in each group. **(F)** Immunoblot quantification of LCAD in each group. **(G)** CPT1 mRNA expression level in the gastrocnemius muscle of rats in each group. **(H)** Relative protein expression of CPT1 in the gastrocnemius muscle of rats in each group. **(I)** Immunoblot quantification of CPT1 in each group. Statistical analyses were performed using the one-way ANOVA followed by LSD post-hoc to compare variances. Data represented mean±standard deviation. $^{*}P<0.05$; $^{**}P<0.01$; $^{***}P<0.001$; $^{****}P<0.0001$; ns, not significant.

## Effects of down-regulation of miR-27a on systolic blood pressure, Lee's index, visceral adipose weight, serum lipids and HOMA-IR in rats

As shown in Fig 4, The changes in systolic blood pressure of rats in each group during the experiment were shown in Panel C. At 32 weeks of age, rats in the MS-S-N ($P<0.01$), MS-S-I ($P<0.0001$), MS-E-N ($P<0.001$), and MS-E-I

**Table 3. MS rat modeling validation.**

| Group | SD group (n = 18) | MS group (n = 18) |
|---|---|---|
| Systolic blood pressure (mmHg) | 108.42 ± 6.41 | 134.95 ± 13.21*** |
| Diastolic blood pressure (mmHg) | 78.66 ± 7.25 | 107.44 ± 12.87*** |
| Fasting blood glucose | 5.49 ± 0.29 | 6.51 ± 0.55*** |
| Body weight (g) | 597.39 ± 61.10 | 638.84 ± 55.73* |
| Lee's index | 3.10 ± 0.06 | 3.13 ± 0.10 |

Statistical analyses were performed using the Student's t-test to compare variances. Data represented mean ± standard deviation. *$P < 0.05$, ***$P < 0.001$ compared to SD group.

($P < 0.0001$) groups had significantly higher systolic blood pressure compared to the SD-S-N group, and all of them were >130 mm Hg, which was in accordance with the modeling criteria. Adeno-associated virus injection carrying miR-27a sponge was performed. At 36 weeks of age, rats in the MS-S-I ($P < 0.01$) and MS-E-I ($P < 0.0001$) groups showed a decreasing trend in systolic blood pressure. Subsequently, 8 weeks of aerobic exercise were carried out, and at 44 weeks of age of the rats, systolic blood pressure was significantly higher in the MS-S-N group than in the SD-S-N group, whereas it showed a different degree of decrease in the MS-E-N, MS-S-I ($P < 0.05$), and MS-E-I ($P < 0.0001$) groups compared with that of the MS-S-N group.

The Lee's index in each group was measured, as shown in Panel D. Compared with the MS-S-N group, the Lee's index of rats in the MS-E-N ($P < 0.05$) and MS-E-I ($P < 0.01$) groups were decreased, indicating that aerobic exercise intervention and silencing miR-27a combined with aerobic exercise were effective in decreasing the Lee's index of MS rats. The Lee's index of rats in the MS-E-I group was also lower compared with that in the MS-S-I group ($P < 0.05$), indicating that the silencing of miR-27a combined with aerobic exercise could further reduce the Lee's index of MS rats compared with aerobic exercise alone.

MS-S-N had the highest fat weights compared to the other groups, with MS-E-I showing a significant decrease in visceral adipose weights compared to the MS-S-N group ($P < 0.05$). Visceral adipose weight was significantly higher in SD-E-I group compared with MS-S-N, MS-S-I, and MS-E-N group ($P < 0.001$, $P < 0.05$, $P < 0.05$), whereas the difference was not statistically significant in SD-E-I compared with MS-E-I (Panel E).

As shown in Panel F-I, HOMA-IR was increased in MS-S-N compared with SD-S-N ($P < 0.05$). Compared with MS-S-N, triglyceride content was lower in MS-S-I ($P < 0.05$), LDL-C was lower in MS-E-I and MS-E-N groups ($P < 0.05$, $P < 0.05$), and HDL-C was higher in MS-E-N group ($P < 0.05$).

## Changes in cell morphology observed by HE staining of rat liver and skeletal muscle tissues

As can be seen from Fig 5, the hepatocytes in liver tissues of rats in the SD-S-N group were arranged in a neat and orderly manner, with uniform sizes, and no abnormal vacuole-like lipid droplets appeared, whereas the hepatocytes in liver tissues of rats in the MS-S-N group were enlarged and disorganized, with the appearance of lipid droplets of varying sizes in vacuoles in the cytoplasm, with nuclei squeezed to the side, and with some of them being necrotic and apoptotic, and with nuclei consolidated, fragmented, and dissolved, with nuclei becoming smaller, and with the cytoplasm becoming darker in coloring. The nuclei became smaller and cytoplasmic staining became darker. Compared with the MS-S-N group, the hepatocytes of rat liver tissues in the MS-E-N, MS-S-I, and MS-E-I groups were more orderly, and although lipid droplet vacuoles existed, they were mostly smaller and fewer (Panel A). Similarly, lipid droplet vacuoles could be seen in skeletal muscle of rats in the MS-S-N group in the cytoplasm of skeletal cells, and the nuclei of the cells had abnormal morphology with lysis and fragmentation. This phenomenon was improved in MS-E-N, MS-S-I, and MS-E-I groups (Panel B).

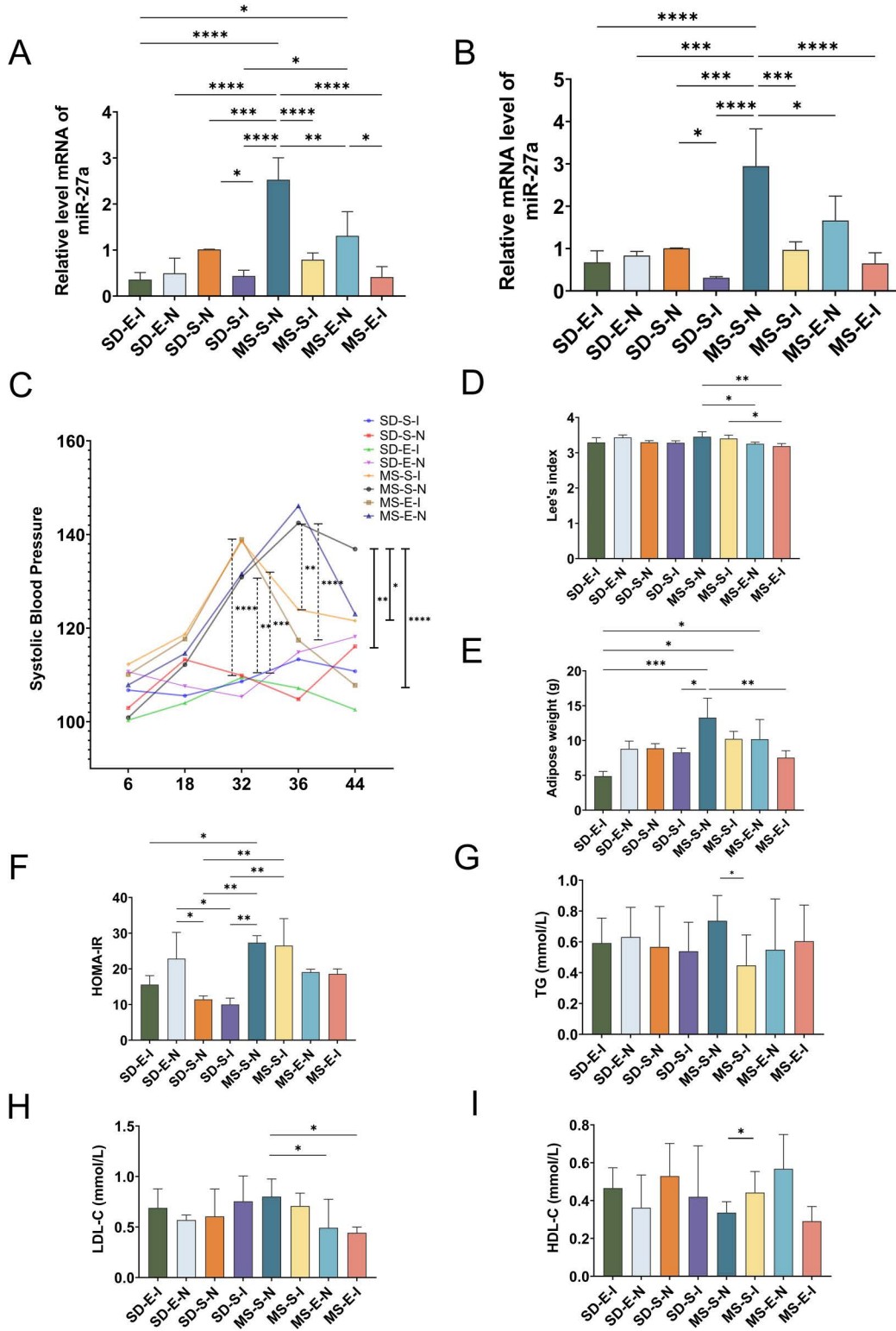

**Fig 4. MiR-27a mRNA expression in liver and skeletal muscles of rats in each group, systolic blood pressure variations, Lee's index, adipose weight, HOMA-IR, and serum lipid levels in rats in each group (MS-E-I: MS rats-Exercise-Inhibition group, n = 4; MS-E-N: MS rats-Exercise-Null**

group, n=4; MS-S-I: MS rats-Sedentary-Inhibition group, n=5; MS-S-N: MS rats-Sedentary-Null group, n=5; SD-E-I: SD rats-Exercise-Inhibition group, n=4; SD-E-N: SD rats-Exercise-Null group, n=4; SD-S-I: SD rats-Sedentary-Inhibition group, n=5; SD-S-N: SD rats-Sedentary-Null group, n=5). **(A)** miR-27a mRNA expression level in liver of rats in each group. **(B)** miR-27a mRNA expression level in skeletal muscle of rats in each group. **(C)** The level of systolic blood pressure in each group of rats from the beginning to the end of the experiment. **(D)** Lee's index of rats in each group at the end of the experiment. **(E)** Adipose weight of rats in each group at the end of the experiment. **(F)** HOMA-IR levels of rats in each group at the end of the experiment. **(G-I)** Serum lipids content of rats in each group. Except for SBP, which was analyzed using two-way analysis of variance, all other indicators were analyzed by using the one-way analysis of variance followed by LSD post-hoc to compare variances. Data represented mean±standard deviation. $^*P<0.05$; $^{**}P<0.01$; $^{***}P<0.001$; $^{****}P<0.0001$.

## mRNA and protein expression of PPARγ, GLUT4, and IRS-1 in rat skeletal muscle after down-regulation of miR-27a

As shown in Fig 6, compared with the SD-S-N group, PPARγ protein expression was significantly decreased in the MS-S-N group ($P<0.001$); compared with the MS-S-N group, expression level of PPARγ mRNA in MS-S-I group ($P<0.05$), PPARγ mRNA and protein in MS-E-N group ($P<0.05$, $P<0.05$), and PPARγ mRNA and protein in MS-E-I group ($P<0.01$, $P<0.01$) were significantly increased. To varying degrees, PPARγ's mRNA or protein expression could be increased by aerobic activity or miR-27a down-regulation. Notably, when aerobic exercise was added after miR-27a was down-regulated, the MS-E-I group showed a greater degree of increase in PPARγ mRNA and protein expression than the MS-S-N group ($P<0.01$, $P<0.01$) (Panel A-D).

As shown in Panel E-H, GLUT4 mRNA and protein expression were lower in the MS-S-N group compared to the SD-S-N group; GLUT4 mRNA and protein expression tended to increase in the MS-S-I and MS-E-N groups compared to the MS-S-N group. The MS-E-I group showed a more notable increase in GLUT4 mRNA ($P<0.05$) and protein ($P<0.001$) expression in comparison to the MS-S-N group, and GLUT4 protein expression was significantly higher in the MS-E-I group compared to the MS-S-I group ($P<0.01$). GLUT4 mRNA or protein expression could be increased to varying degrees by aerobic exercise, miR-27a downregulation and miR-27a downregulation combined with aerobic exercise.

As shown in Panel I-L, when compared to the SD-S-N group, IRS-1 protein expression was lower in the MS-S-N group ($P<0.05$). However, there was an increasing trend in mRNA and protein expression in the MS-S-I and MS-E-N groups when compared to the MS-S-N group. Additionally, after down-regulating miR-27a, increased aerobic exercise was observed in the MS-E-I group, which resulted in a significant increase in mRNA and protein expression ($P<0.01$, $P<0.05$). Within the SD rats, there was a significant increase in IRS-1 protein expression in the SD-E-I group when compared to the SD-E-N, SD-S-N, SD-S-I ($P<0.01$, $P<0.01$, $P<0.01$), and MS-S-N, MS-S-I, MS-E-N, MS-E-I group ($P<0.0001$, $P<0.0001$, $P<0.0001$, $P<0.001$).

## mRNA and protein expression of PPARγ, LCAD, and CPT1 in rat liver after down-regulation of miR-27a

As can be seen from Fig 7, PPARγ mRNA and protein expression were lower in the MS-S-N group compared with the SD-S-N group; PPARγ mRNA expression in the MS-S-I group was increased compared with the MS-S-N group ($P<0.05$), and PPARγ mRNA and protein expression were significantly increased in the MS-E-I group ($P<0.001$, $P<0.05$). This suggests that down-regulation of miR-27a followed by increased aerobic exercise more significantly increased hepatic PPARγ expression. Compared with the SD-S-N group, PPARγ mRNA expression was increased in the SD-S-I ($P<0.05$), SD-E-N, and SD-E-I ($P<0.01$) groups (Panel A-D).

As can be seen from Panel E-H, LCAD expression was lower in the MS-S-N group than in the SD-S-N group; in contrast to the MS-S-N group, LCAD mRNA expression in the MS-S-I ($P<0.05$) and MS-E-I ($P<0.01$) groups showed a trend of increasing; in the MS-E-I group, LCAD protein expression was significantly increased ($P<0.01$), and in the MS-E-I group, LCAD protein expression was significantly higher ($P<0.05$) than in the MS-E-N group.

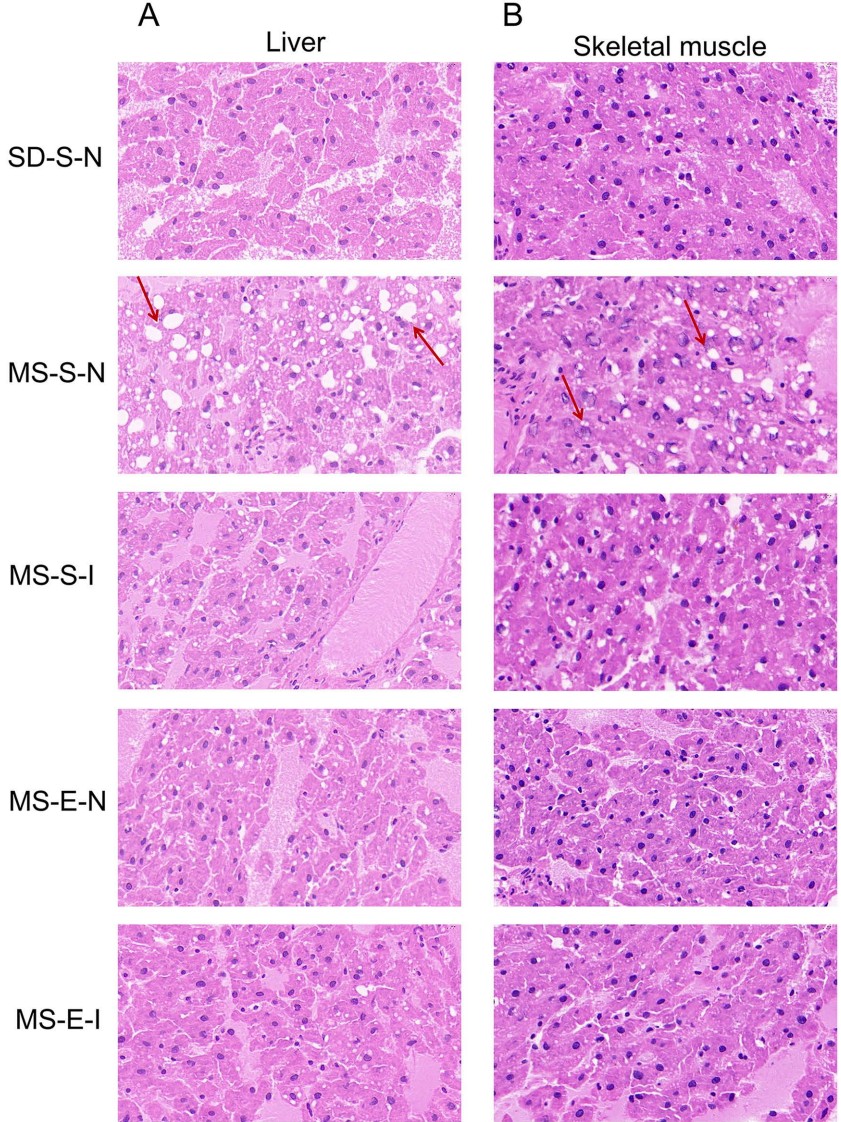

**Fig 5. HE staining results of rat liver and skeletal muscle. (A)** HE staining of liver in SD-S-N, MS-S-N, MS-S-I, MS-E-N, and MS-E-I groups (MS-E-I: MS rats-Exercise-Inhibition group, n = 4; MS-E-N: MS rats-Exercise-Null group, n = 4; MS-S-I: MS rats-Sedentary-Inhibition group, n = 5; MS-S-N: MS rats-Sedentary-Null group, n = 5; SD-S-N: SD rats-Sedentary-Null group, n = 5). **(B)** HE staining of skeletal muscle in SD-S-N, MS-S-N, MS-S-I, MS-E-N, and MS-E-I groups.

As can be seen from Panel I-L, the MS-S-N group showed a decrease in CPT1 protein expression ($P < 0.01$) when compared to the SD-S-N group. On the other hand, the CPT1 mRNA and protein expression in the MS-S-I ($P < 0.05$, $P < 0.01$) and MS-E-I ($P < 0.001$, $P < 0.05$) groups showed an increase when compared to the MS-S-N group. Furthermore, the mRNA expression of MS-E-I was found to be much higher than that of MS-S-I ($P < 0.001$), suggesting that the addition of aerobic exercise, which was based on the down-regulation of miR-27a, was more successful in up-regulating the expression of CPT1. In comparison to the SD-S-N group, there was an increase in CPT1 mRNA expression in the SD-E-I group ($P < 0.001$).

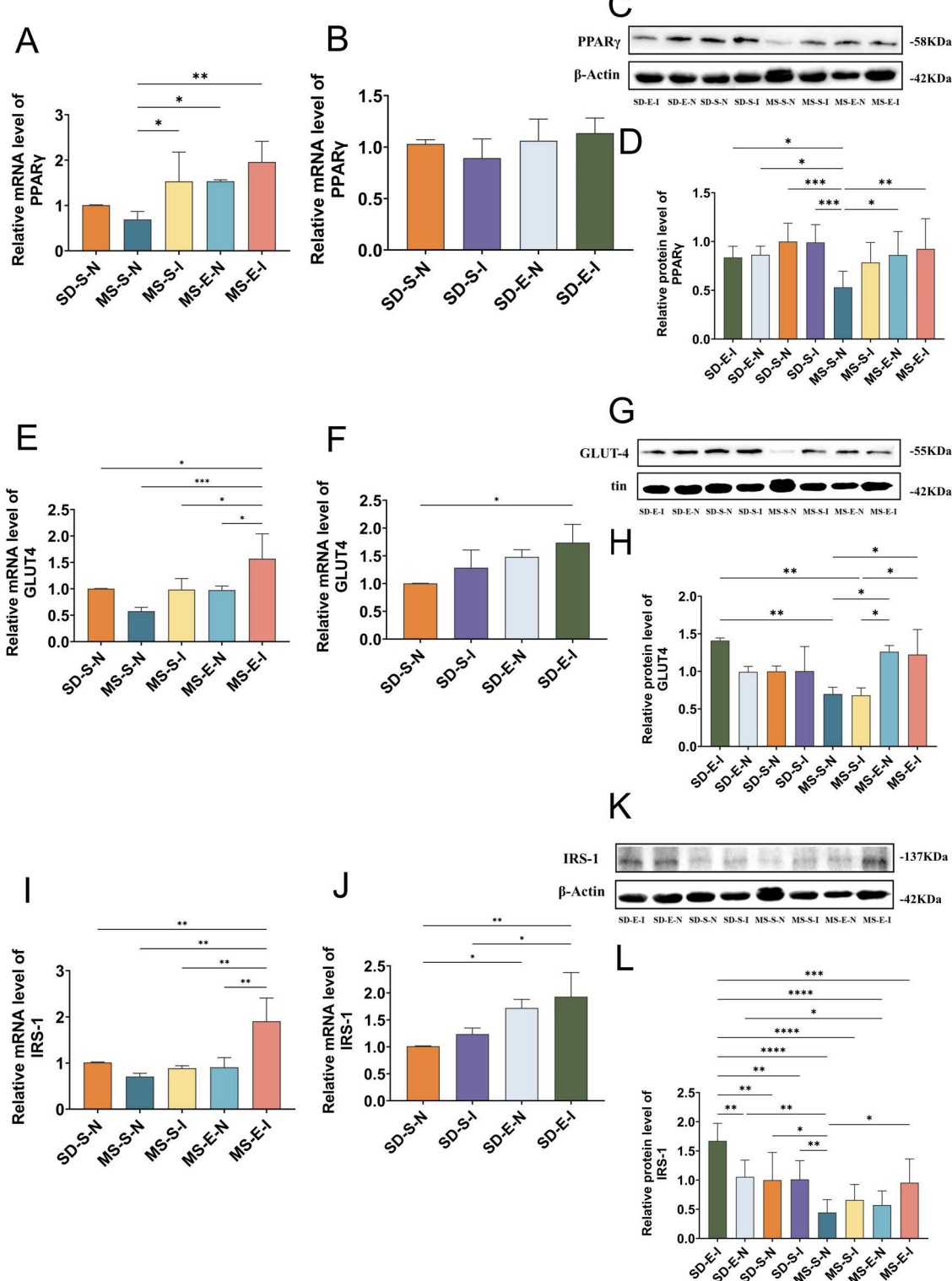

**Fig 6. mRNA and protein expression of PPARγ, GLUT4, and IRS-1 in rat skeletal muscle after down-regulation of miR-27a (MS-E-I: MS rats-Exercise-Inhibition group, n = 4; MS-E-N: MS rats-Exercise-Null group, n = 4; MS-S-I: MS rats-Sedentary-Inhibition group, n = 5; MS-S-N:**

MS rats-Sedentary-Null group, n=5; SD-E-I: SD rats-Exercise-Inhibition group, n=4; SD-E-N: SD rats-Exercise-Null group, n=4; SD-S-I: SD rats-Sedentary-Inhibition group, n=5; SD-S-N: SD rats-Sedentary-Null group, n=5). (A, B) Quantitative comparison of PPARγ mRNA level in different groups. (C, D) The relative protein expression level of PPARγ and immunoblotting band in each group. (E, F) Quantitative comparison of GLUT4 mRNA level in different groups. (G, H) The relative protein expression level of GLUT4 and immunoblotting band in each group. (I, J) Quantitative comparison of IRS-1 mRNA level in different groups. (K, L) The relative protein expression level of IRS-1 and immunoblotting band in each group. Statistical analyses were performed using the one-way ANOVA followed by LSD post-hoc to compare variances. Data represented mean±standard deviation. $^*P<0.05$; $^{**}P<0.01$; $^{***}P<0.001$; $^{****}P<0.0001$.

## Discussion

MS has been a serious hazard to human health in recent decades and is closely associated with the onset of diabetes and cardiovascular diseases [23]. Numerous studies have shown that aerobic exercise is an effective method for preventing and treating MS [5]. Nevertheless, the molecular mechanisms underlying the effects of aerobic exercise on metabolic illnesses remain unknown. In order to clarify the actual effect of aerobic exercise to improve the disorders of glycolipid metabolism in MS rats and further verify the pathway of its mechanism of action, the study developed an MS rat model to mimic the process of underexercise and overnutrition, followed by an 8-week aerobic exercise regimen. This study's key discovery was that rats with MS were caused to have abnormal glucose-lipid metabolism as a result of eating a diet high in fat and sugar. Consistent with previous findings [24–26]. we showed that aerobic exercise exerted an influential role in weight loss, improved insulin resistance and ameliorated lipid metabolism during the development of MS in rats, and these effects were related to the decreased expression of miR-27a and increased expression of PPARγ.

MS is typically characterized by obesity, hypertension and insulin resistance [1]. The present study firstly focused on characterizing the physical and metabolic changes that occur as a result of high-fat and high-sugar dietary intake compared to animals fed a standard diet, and showed the role of aerobic exercise in mitigating such changes. Lee's index is an effective index in evaluating the degree of obesity in rats [21], and the HOMA-IR is a preliminary indicator to determine the degree of insulin resistance, indicating the the relationship between glucose and insulin kinetics during fasting [27]. Consequently, we used Lee's index to assess the degree of obesity in rats, and the HOMA-IR formula to initially assess the level of insulin resistance in this study. Both systolic blood pressure and Lee's index were significantly elevated in MS rats, whereas MS rats that underwent treadmill running exercise exhibited improved systolic blood pressure and Lee's index, suggesting that aerobic exercise slowed down the significant increase in blood pressure and obesity associated with a high-fat and high-sugar diet. This was consistent with the findings of Er.F et al. that aerobic exercise was effective in decreasing systolic blood pressure and Lee's index in MS rats [28].The HOMA-IR result suggested that a high-fat and high-sugar diet contributed to significant insulin resistance in rats, whereas aerobic exercise relatively attenuated the effects of the high-fat and high-sugar diet. Thus, aerobic exercise was able to reduce systolic blood pressure, Lee's index and insulin resistance in MS rats.

Overexpression of miR-27a is associated with disorders of glucolipid metabolism, cardiovascular disease, and bone disorders, whereas PPARγ affects energy balance, lipid metabolism, and insulin-stimulated glucose uptake [8,29,30]. Our study proposed that a high-fat and high-sugar diet may inhibit PPARγ expression by inducing miR-27a expression, promoting insulin resistance and disrupting lipid metabolism, and aerobic exercise reversing these changes. In our team's previous experiments, 8 weeks of treadmill running was found to significantly induce PPARs mRNA expression in liver, adipose tissue, and skeletal muscle of MS rats [16]. By blocking PPARγ-induced adipocyte differentiation and insulin resistance, miR-27a may be a major pathogenic mechanism causing insulin resistance and dyslipidemia in MS [13,14]. Notably, clinical studies have also identified significantly elevated levels of miR-27a in the serum of patients with MS and type 2 diabetes, further supporting its potential role as a key regulatory factor in metabolic disorders [31,32]. MiRNAs can specifically recognize the corresponding target sites on the 3'UTR of target genes and inhibit the expression of the target genes by base complement pairing and binding to them. and inhibit the expression of target genes through base

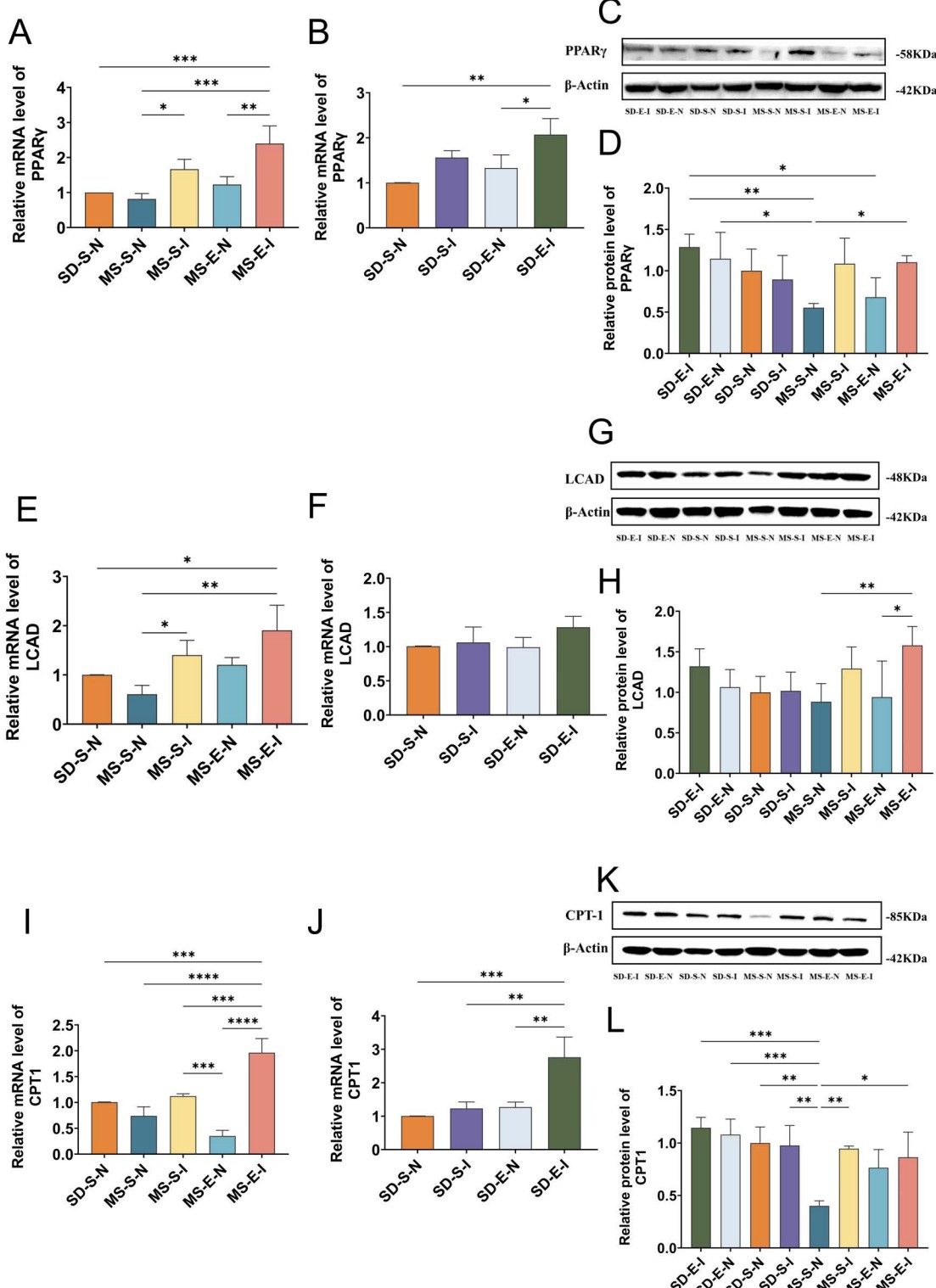

**Fig 7. mRNA and protein expression of PPARγ, LCAD, and CPT1 in rat liver after down-regulation of miR-27a (MS-E-I: MS rats-Exercise-Inhibition group, n = 4; MS-E-N: MS rats-Exercise-Null group, n = 4; MS-S-I: MS rats-Sedentary-Inhibition group, n = 5; MS-S-N: MS rats-Sedentary-Null group, n = 5; SD-E-I: SD rats-Exercise-Inhibition group, n = 4; SD-E-N: SD rats-Exercise-Null group, n = 4; SD-S-I: SD**

rats-Sedentary-Inhibition group, n = 5; SD-S-N: SD rats-Sedentary-Null group, n = 5). **(A, B)** Quantitative comparison of PPARγ mRNA levels in each group. **(C, D)** The relative protein expression level of PPARγ and immunoblotting band in each group. **(E, F)** Quantitative comparison of LCAD mRNA levels in each group. **(G, H)** The relative protein expression level of LCAD and immunoblotting band in each group. **(I, J)** Quantitative comparison of CPT1 mRNA levels in each group. **(K, L)** The relative protein expression level of CPT1 and immunoblotting banding each group. Statistical analyses were performed using the one-way ANOVA followed by LSD post-hoc to compare variances. Data represented mean ± standard deviation. $^{*}P < 0.05$; $^{**}P < 0.01$; $^{***}P < 0.001$; $^{****}P < 0.0001$.

complement pairing [33]. The gene prediction database predicted the downstream target genes of miR-27a and found that the protein could bind to the positional sequence (ACUGUGAA) at bases 88–95 in the coding region of the rat PPARγ gene sequence. These findings are consistent with the results of our study: by observing the changes in miR-27a and PPARγ expression, we discovered that a high-fat and high-sugar diet elevated miR-27a expression and decreased PPARγ expression, and the trend toward increased PPARγ expression in skeletal muscle and liver of rats was negatively correlated with miR-27a expression. Furthermore, after 8 wk of aerobic exercise, whether in skeletal muscle or liver, miR-27a expression was lower in the MS-E group than in the MS-S group, and PPARγ expression was greater than that in the MS-S group, demonstrating that aerobic exercise training may promote increased expression of PPARγ by inhibiting miR-27a. Additionally, a dual-luciferase assay in this study demonstrated a targeting relationship between miR-27a and PPARγ and suggested that miR-27a could potentially negatively regulate the expression of PPARγ. This was in line with those of Chen T and Liu J [34,35]. Consequently, increase of PPARγ and its downstream target gene expression may occur in conjunction with downregulation of miR-27a expression. A high-fat and high-sugar diet may inhibit PPARγ expression by inducing miR-27a expression, promoting insulin resistance and disrupting lipid metabolism, and aerobic exercise reversing these changes.

Although our findings indicated that aerobic exercise up-regulated PPARγ by reducing miR-27a levels, the exact mechanism through which exercise exerted this control remained to be elucidated. Recent work had shown that exercise modulated exosome release, thereby facilitating the exchange of proteins, miRNAs, mRNAs and mitochondrial DNA among cells and tissues, and consequently contributed to the regulation of gene expression and systemic physiology [36,37]. We therefore hypothesised that aerobic exercise had promoted the sorting of miR-27a into exosomes and its subsequent secretion into the circulation, accelerating the clearance of this miRNA from insulin-sensitive tissues such as skeletal muscle and liver. Such an exosome-based transport mechanism could represent an important post-transcriptional response to physiological stimuli. To test this hypothesis, follow-up studies should profile the miRNA cargo of exosomes isolated from blood and tissues after exercise, and intervene in exosome biogenesis or secretion, so as to clarify the upstream events through which exercise influenced miR-27a abundance.

Next, to investigate the functional consequences of PPARγ upregulation, we examined the expression of its downstream target genes associated with glucose and lipid metabolism. Skeletal muscle is crucial for exercise, thermoregulation, and glucose homeostasis [38], and is a key component of the response to insulin-induced glucose uptake processes and a key location of peripheral insulin resistance [39,40]. GLUT4 and IRS-1 are crucial signaling molecule in the insulin signaling pathways that react to glucose transport and mediate insulin signaling [38,41]. A significant metabolic change linked to obesity and type 2 diabetes is insulin resistance in skeletal muscle, which affects blood glucose levels and leads to altered insulin signaling throughout the body [42]. Thus, insulin signaling exerts an impact on glucose metabolism in skeletal muscle. By modifying the signaling pathways involved in glucose uptake, exercise can help treat abnormalities in glucose metabolism [43]. Therefore, we determined the expression of receptors related to glucose metabolism downstream of PPARγ in skeletal muscle and found that a high-fat and high-sugar diet decreased the expression of GLUT4 and IRS-1 at both the mRNA and protein levels, and aerobic exercise ameliorated this reduction in expression. Insulin primarily enhances glucose uptake in skeletal muscle by increasing microvascular blood flow, completing transport and increasing GLUT4 translocation [44]. The most abundant glucose transporter protein isoform expressed in skeletal muscle is GLUT4

[38]. For skeletal muscle to absorb glucose more effectively, GLUT4 must be expressed at a higher level on the surface of the myocytes. The results of the immunohistochemistry studies that we conducted to assess the expression of GLUT4 in skeletal muscle were in agreement with those of the RT–qPCR and WB experiments. The positive expression of GLUT4 in skeletal muscle was reduced in the MS-S group, while aerobic exercise effectively alleviated this downregulation, and the expression of GLUT4 was significantly greater in MS-E rats than in MS-S rats.

The liver, on the other hand, is the largest metabolic organ in the body; it secretes triglycerides as lipoproteins and directs lipids to oxidative or esterification pathways through the intake and production of fatty acids [45–47]. Lipid accumulation in liver and impaired insulin signaling affect the ratio of hepatic triglyceride stores to lipoprotein secretion, and lipid homeostasis in liver is dependent on lipid efflux, so that systemic metabolism of lipoproteins is closely linked to lipid homeostasis in liver [48–50]. Changes in serum triglycerides and lipoproteins in MS rats suggested an impaired capacity for liver lipid homeostasis. The MS rats had aggregated triglycerides and accordingly exhibited higher LDL-C levels and lower HDL-C levels in serum compared with the SD rats. However, aerobic exercise improved the elevated serum triglycerides and LDL-C and increased HDL-C. This was consistent with other studies that have shown that regular aerobic exercise increases HDL-C and decreases LDL-C and triglyceride levels [51]. In addition to lowering serum triglycerides and improving lipoprotein secretion, exercise reduces high-fat diet-induced hepatic triglyceride levels by modulating fatty acid β-oxidation and upregulating PPARγ to improve lipid deposition [52,53]. Therefore, changes in the expression of relevant enzymes involved in fatty acid β-oxidation may be indicative of hepatic lipid-processing capacity, and in the present study, we examined the expression of important enzymes associated with PPARγ downstream lipid metabolism in liver. The rate-limiting enzyme for the transport of fatty acids to mitochondria to complete β-oxidation is called CPT1, and it also has a major function [54]. Energy from fatty acid metabolism is obtained through the oxidation of long-chain fatty acids in the mitochondria, mainly through dehydrogenation assisted by LCAD [55]. In this study, the results showed that a high-fat and high-sugar diet reduced the mRNA and protein expression of LCAD and CPT1, while aerobic exercise counteracted this reduction in expression, and improved lipid processing capacity in liver. Thus, aerobic exercise reduced serum triglyceride levels, improved lipoprotein secretion, and increased the expression of important enzymes related to lipid metabolism downstream of PPARγ in liver.

This study's first section established that aerobic exercise was beneficial for lowering blood pressure, weight loss, and increasing glycolipid metabolism in MS rats. Aerobic exercise had been shown to down-regulate the expression of miR-27a and up-regulate the expression of PPARγ, GLUT4, IRS-1, LCAD, and CPT1. However, more research is needed to determine the critical role that the miR-27a-PPARγ signaling pathway plays in mitigating disorders of glycolipid metabolism and to understand the relationships that exist between regulating the expression of proteins and receptors involved in glucose metabolism, the expression of lipid metabolism-related enzymes, and the expression of miR-27a and PPARγ.

Although there is currently a dearth of comprehensive and sufficient study on the role of abnormal alterations in miRNA in metabolic syndrome, it is possible that these changes play a significant role in the development of the condition. In the present study, starting from the role of miR-27a in regulating lipid metabolism, insulin sensitivity, etc., we explored whether miR-27a is crucial in the onset of MS by injecting rats with adeno-associated viruses that carry miR-27a sponges to suppress the expression of miR-27a in bloodstreams. Four weeks after the injection of adeno-associated virus, an eight-week rat treadmill exercise was carried out to further investigate whether aerobic exercise combined with miR-27a down-regulation could further improve the disorders of glucose-lipid metabolism in MS rats compared with down-regulation of miR-27a alone, so as to clarify whether miR-27a plays a key role in the process of aerobic exercise to improve MS.

Biochemical changes observed in MS, including dysregulation of glucose and lipid metabolism, immune responses, and endothelial cell dysfunction, serve as a pathological bridge between the promotion of metabolic syndrome, diabetes mellitus, and cardiovascular and neurodegenerative diseases [56]. The results of the second section showed that in MS rats, down-regulating miR-27a improved serum lipids, visceral fat weight, systolic blood pressure, and HOMA-IR. Following the aerobic exercise, the MS-S-I group displayed a decreasing trend in visceral fat weight when compared

to the MS-S-N group. Additionally, the systolic blood pressure showed a significant down-regulation, and the contents of triglyceride, LDL-C, and HDL-C increased. This trend was more pronounced in the MS-E-I group, and a decrease in HOMA-IR was also observed. This indicates that in MS rats, aerobic exercise combined with miR-27a down-regulation can more effectively reduce blood glucose, attenuate visceral fat formation, alleviate dyslipidemia, and reduce hypertension than miR-27a down-regulation alone.

The mechanism of lipid-induced insulin resistance is revealed by a study published in Lancet: intracellular lipid buildup in liver and muscle activates a new protein kinase C, which in turn impairs insulin signaling [57]. In this study, the morphology of skeletal muscle and liver tissues was observed by HE staining for the presence of abnormal lipid accumulation. The results showed that compared with the control group, the cytoplasm of the skeletal muscle and liver tissues of rats in the MS-S-N group showed lipid droplet vacuoles of different sizes, disorganized arrangement of nuclei, and irregular morphology, which was partially improved in the MS-E-N and MS-S-I groups, and further improved in the MS-E-I group. Accumulation of lipid-rich lipid droplets in liver is a hallmark of nonalcoholic fatty liver disease and is directly linked to malfunction of lipid metabolism. These conditions can substantially compromise normal liver function and cause long-term damage to the liver tissue [58,59]. Additionally, there is mounting evidence linking decreased insulin sensitivity in skeletal muscle to lipid deposition in the myocytes of the muscle. Extra fat is esterified, stored, or converted into a variety of compounds that can either support or disrupt regular cellular activity, especially insulin-mediated signaling. Defects in insulin-induced glucose transport activity cause lipid-induced insulin resistance in skeletal muscle [60]. HE staining results suggested that after aerobic exercise or down-regulation of miR-27a expression, the catabolic rate of rats is greater than the anabolic rate, which can improve the distribution of lipid droplets in skeletal muscle and liver tissue cells, thus normalizing the cellular morphology, and has a protective effect on skeletal muscle and liver tissue cells, which is in agreement with the findings of Deng K [61]. This suggested that miR-27a down-regulation followed by intervening exercise improves liver and skeletal muscle lipid accumulation induced by high-fat and high-sugar diet feeding in MS rats.

We observed that in skeletal muscle tissues, mRNA expression of PPARγ was increased after down-regulation of miR-27a, and the mRNA and protein of downstream GLUT4 and IRS-1 showed a trend of increase; similarly, in liver tissues, the mRNA expression of PPARγ was also increased, and the mRNA of LCAD and the protein of CPT1 of the downstream lipid cycle-associated enzyme were increased compared with the MS-S-N group, but without statistical significance. This may be due to the fact that adeno-associated viruses carrying the miR-27a sponge mostly reached peak expression in rats after 4 weeks of injection, but the expression level of adeno-associated viruses gradually decreased with the extension of time. At the end of the experiment, this inhibition of miR-27a expression gradually weakened, and the continued feeding of high-fat and high-sugar chow may have attenuated this ameliorating effect. However, in the group that underwent aerobic exercise intervention after down-regulation of miR-27a, the mRNA and protein expression of PPARγ was significantly increased, and the mRNA and protein of downstream GLUT4 and IRS-1 also showed significant increases in skeletal muscle tissues. In liver tissues, the mRNA and protein expression of PPARγ was significantly increased, and the mRNA and protein of downstream LCAD and CPT1 were all significantly increased. This suggests that the effect achieved by silencing miR-27a was facilitated after intervening exercise. Consistent with earlier findings that adipocyte differentiation, cardiovascular dysfunction, and insulin resistance could be improved by down-regulation of miR-27a [62–64]. Therefore, down-regulation of miR-27a followed by aerobic exercise intervention helped to mitigate these degenerative alterations linked to glycolipid metabolism. This suggested that low expression of miR-27a has a very critical role in the improvement of glucose metabolism and lipid metabolism in MS rats by aerobic exercise. Some of the expected effects did not approach statistical significance, which could be explained by sample size restrictions. With larger sample sizes, these effects may become more pronounced, providing stronger evidence to support our hypotheses.

Thus, the results of the second part of this study suggest that down-regulation of miR-27a partially ameliorates obesity, hypertension, and dyslipidemia in MS rats and differentially increases the protein and mRNA expression of PPARγ, GLUT4, and IRS-1 in skeletal muscle and PPARγ, LCAD, and CPT1 in liver. This ameliorative effect was further enhanced

after the intervention of aerobic exercise, suggesting that miR-27a plays a key role in the ameliorative effects of aerobic exercise-mediated MS.

A key strength of this study is the translation of previous in vitro results to an in vivo setting, confirming PPARγ upregulation-beyond IRS-1 inhibition-as a central mechanism, as reported by Yu et al. [14], and validating the regulatory function of miR-27a under physiological conditions, consistent with Chen T et al. [34]. However, several limitations must be acknowledged. First, we did not examine whether miR-27a suppression improves inflammation or fibrosis in MS, which are critical to disease progression. Second, potential synergistic effects involving other miRNAs in the regulatory network cannot be ruled out. Third, although effect sizes were observed for primary outcomes, the sample size may limit power to detect smaller effects or complex interactions, increasing the risk of Type II errors. Consequently, caution is warranted when interpreting negative outcomes, and larger studies are needed to confirm these findings. Future research should use single-cell sequencing to clarify tissue-specific mechanisms and explore combined exercise-genetic strategies for improved clinical translation.

## Conclusion

Aerobic exercise effectively reduced blood pressure, alleviated obesity, improved insulin resistance and dyslipidemia. It inhibited the expression of miR-27a in the skeletal muscle and liver of MS rats, upregulated the expression of PPARγ and downstream signaling molecules and enzymes related to glucose and lipid metabolism, and thus improved the disorder of glucose and lipid metabolism. In MS rats, miR-27a mediated its negative regulatory effect by binding to the partial coding region of PPARγ, promoting the occurrence of glucose and lipid metabolism disorders. Aerobic exercise exerted positive effects in reducing lipid and weight, and lowering blood pressure by regulating the miR-27a-PPARγ pathway, effectively improving the disorder of glucose and lipid metabolism.

## Author contributions

**Conceptualization:** Zhizhuo Wang, Yao Gao, Jinwu Wang, Chunlei Shan, Qi Guo, Cheng Lin.

**Data curation:** Peiyun Wu, Kunhui Li, Yao Gao, Han Yang, Cheng Lin.

**Formal analysis:** Peiyun Wu, Kunhui Li, Qi Guo.

**Funding acquisition:** Jinwu Wang, Chunlei Shan, Qi Guo, Cheng Lin.

**Investigation:** Yao Gao, Chunlei Shan, Han Yang, Cheng Lin.

**Methodology:** Peiyun Wu, Zhizhuo Wang, Kunhui Li, Jing Yang, Chunlei Shan, Yuanshan Jiang, Cheng Lin.

**Project administration:** Zhizhuo Wang, Cheng Lin.

**Resources:** Yao Gao, Jinwu Wang, Han Yang.

**Software:** Yao Gao.

**Supervision:** Zhizhuo Wang, Cheng Lin.

**Validation:** Peiyun Wu, Kunhui Li, Jing Yang, Juan Wang, Yuanshan Jiang.

**Visualization:** Peiyun Wu, Kunhui Li, Jing Yang, Juan Wang.

**Writing – original draft:** Peiyun Wu, Zhizhuo Wang, Jing Yang, Cheng Lin.

**Writing – review & editing:** Peiyun Wu, Zhizhuo Wang, Cheng Lin.

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
