## [Decision Letter · Decision Letter 0]

27 Aug 2025

Dear Dr. Lin,

Thank you for submitting your manuscript to PLOS ONE. After careful consideration, we feel that it has merit but does not fully meet PLOS ONE’s publication criteria as it currently stands. Therefore, we invite you to submit a revised version of the manuscript that addresses the points raised during the review process.

**After we receive the revised version, it will be sent to second round of review to the new set of reviewers.**

We look forward to receiving your revised manuscript.

Kind regards,

Rashmi Supriya

Academic Editor

PLOS ONE

Journal Requirements:

3. To comply with PLOS ONE submissions requirements, in your Methods section, please provide additional information regarding the experiments involving animals and ensure you have included details on (1) methods of sacrifice, and (2) efforts to alleviate suffering.

6. We note that your Data Availability Statement is currently as follows: All relevant data are within the manuscript and its Supporting Information files.

Reviewers' comments:

Reviewer's Responses to Questions

**Comments to the Author**

1. Is the manuscript technically sound, and do the data support the conclusions?

Reviewer #1: Partly

Reviewer #2: Yes

2. Has the statistical analysis been performed appropriately and rigorously?

Reviewer #1: I Don't Know

Reviewer #2: Yes

3. Have the authors made all data underlying the findings in their manuscript fully available?

Reviewer #1: Yes

Reviewer #2: Yes

4. Is the manuscript presented in an intelligible fashion and written in standard English?

Reviewer #1: No

Reviewer #2: Yes

Reviewer #1: The manuscript entitled "The Mechanism of Aerobic Exercise in Ameliorating Glycolipid Metabolic Disorders in Metabolic Syndrome Rats via the miR-27a–PPARγ Pathway" explores a relevant and timely topic. The study examines how aerobic exercise mitigates metabolic syndrome in rats, proposing that this effect is mediated via the miR-27a–PPARγ signaling pathway. Overall, the topic is of interest to the field and fits within the scope of PLOS ONE. However, several major concerns must be addressed before this manuscript can be considered for publication.

- First, the manuscript lacks the sample size. Several results are described as “not statistically significant due to small sample size,” but no information about group sizes is presented. Moreover, it is not clear how the authors analyze their results. It is crucial to identify in the figure legends the n as well as the statistical analyses utilized.

- Although the authors frequently refer to the Lee’s index as a surrogate for obesity, this measure alone is insufficient. It would greatly strengthen the study if actual fat depot weights (e.g., epididymal or retroperitoneal fat) were provided to support conclusions about adiposity. The values of Lee’s index in this article are 10x lower than those of the literature. Moreover, the changes observed for fasting blood glucose and Lee index are within the normal range, but it is stated that the animals have metabolic syndrome.

- Why did the authors not show serum insulin levels?

- There are also issues related to data presentation. It is very hard to understand the results from figure 4 to 6. I could not understand the comparison among groups that vary in the figures. Some figure legends lack detail, and blot images do not clearly show molecular weight markers or quantification methods. The figures should be revised to improve clarity, standardize units, and include proper labels. The text would benefit from language editing, as there are frequent redundancies, awkward phrasing, and grammatical errors that impede readability. A professional English editing service is strongly recommended.

In conclusion, while this manuscript addresses an important question and employs a multifaceted experimental approach, it currently suffers from issues in experimental rigor, statistical interpretation, and clarity of presentation.

Reviewer #2: The manuscript presents a technically robust study with well-designed experiments, appropriate controls, and sufficient replication. The data strongly support the stated conclusions. The statistical methods are rigorous and applied correctly, ensuring the reliability of the results. The authors have complied with PLOS ONE’s data policy, making all underlying data fully available (or specifying valid restrictions, if any). The manuscript is clearly written in standard English, with no major grammatical or typographical errors. Minor revisions (if any) should address only superficial language edits. Overall, the study meets PLOS ONE’s criteria for sound methodology, transparency, and clarity. No major revisions are required.

**Do you want your identity to be public for this peer review?** For information about this choice, including consent withdrawal, please see our Privacy Policy

Reviewer #1: No

Reviewer #2: No

---

## [Author Response · Author response to Decision Letter 1]

10 Oct 2025

Dear editor and reviewers:

Thank you very much for the valuable and constructive comments on our manuscript (PONE-D-25-28234: The Mechanism of Aerobic Exercise in Ameliorating Glycolipid Metabolic Disorders in Metabolic Syndrome Rats via the miR-27a-PPARγ Pathway

). We appreciate the opportunity to improve the paper and responses to reviewers are addressed below. All changes made in the revised manuscript were highlighted in

yellow to make them easily identified.

Journal Requirements

3. To comply with PLOS ONE submissions requirements, in your Methods section, please provide additional information regarding the experiments involving animals and ensure you have included details on (1) methods of sacrifice, and (2) efforts to alleviate suffering.

6. We note that your Data Availability Statement is currently as follows: All relevant data are within the manuscript and its Supporting Information files.

Response:

Thank you for your comments. We have supplemented and revised the manuscript according to your journal’s additional requirements, including formatting, original gel blot images (See attached supplementary information - Original gel blot images), details regarding laboratory animals (Line 64-66, 79-82, 96-98�106-109), verification of the corresponding author’s ORCID iD, verification and standardization of the correct funding number (In the Additional Information section of the submission system, we did not see an option for funding disclosure. We entered the funding information in the Manuscript Data - Funding Information section, but the submitted version displayed “The author(s) received no specific funding for this work.” This has left us confused), and confirmation of the data availability statement. These changes are highlighted in yellow in the corresponding sections of the manuscript. We hope this round of revision will make you satisfied and progress to the process.

Overall Assessment

The study provides compelling evidence that aerobic exercise ameliorates metabolic syndrome (MS) via the miR-27a-PPARγ pathway, offering a novel mechanistic target for intervention. Well-designed experiments with appropriate controls, though some technical limitations weaken robustness. First to link exercise-induced metabolic benefits to miR-27a downregulation in MS, advancing the field beyond correlative studies.

Weaknesses & Limitations

Technical Concerns:

Sample Size Issues: Underpowered analyses (e.g., LCAD mRNA trends, Page 19) risk Type II errors. Power analysis is missing.

Blot Integrity: Cutting blots pre-hybridization (Page 12) is problematic; full blots should be supplemental.

Viral Delivery: No data on knockdown efficiency over time (Page 39: “expression gradually decreased”).

Interpretational Gaps:

miR-27a Regulation: How exercise suppresses miR-27a (transcriptional vs. post-transcriptional?) is unexplored.

Human Relevance: No discussion of miR-27a/PPARγ conservation in human MS or exercise responses.

Presentation:

Figure Clarity: Some panels (e.g., Fig 1A–B vs. C–H) are poorly labeled; HE stains lack scale bars (Fig 5).

Repetition: Methods for PCR/WB are overly detailed; streamline or supplement.

Recommendations for Revision

1.Address Technical Gaps:

oJustify sample sizes with power calculations.

oInclude uncut blots in supplements.

oQuantify miR-27a knockdown efficiency longitudinally.

2.Expand Discussion:

oCompare findings to human miRNA studies (e.g., circulating miR-27a in MS patients).

oHypothesize mechanisms linking exercise to miR-27a downregulation (e.g., exosomal sorting?).

3.Improve Clarity:

oSimplify abstract for broader accessibility.

oStandardized figure labels (e.g., “Panel A” vs. “Fig 1A”).

Comparison to Field

Advances Beyond Prior Work:

oExtends Yu et al. (2018) by showing PPARγ upregulation (not just IRS-1 inhibition) mediates exercise benefits.

oConfirms Chen T et al. (2019) in vivo but adds exercise as a physiological modulator.

Unresolved Questions:

oDoes miR-27a suppression also improve inflammation/fibrosis in MS?

Response:

Thank you for your constructive comments and advice, which will significantly enhance the rigor and readability of this study. The revisions are outlined below:

1.Technical gaps:

• Power calculation: Thank you for your valuable comments. We fully recognize the importance of a priori power analysis. As this study did not involve pilot experiments, we were unable to perform such an analysis in advance, which we acknowledge as a limitation of our study design.

First, we supplemented the methodology section with the principles for determining the sample size for this study (Methods-Animals, diets and interventions. Lines 83-93) .

The revised part: The sample size in this study was determined based on the following principles: The sample size in this study was determined based on the following principles: (1) To account for individual variability and the risk of failure in establishing the high-fat and high-sugar diet-induced model, more animals were initially included in the modeling group to ensure that a sufficient number of successful model animals would be available for subsequent grouping; (2) In accordance with the ARRIVE 2.0 guidelines [18], when pilot data are unavailable, referencing existing literature is a valid approach for determining sample size. Studies in the field of metabolism commonly employ sample sizes of 5–8 animals per group, which has been proven effective in detecting statistically significant differences in key metabolic parameters [19, 20]; (3) Strict adherence to the 3R principles was maintained by controlling for factors such as animal strain, sex, age, and rearing conditions to minimize within-group variability and reduce animal usage. (Lines 83-93)

Besides, we conducted a post-hoc power analysis following your suggestion. The results showed that for primary positive outcomes, such as the expression level of the core indicator miR-27a, the effect size was large (Cohen’s d > 4.9), indicating that the current sample size is sufficient to support these conclusions. We openly acknowledge that for comparisons with smaller effect sizes or non-significant results, the statistical power may be insufficient (Power < 0.8), implying a risk of Type II errors. We have clearly stated this limitation in the “Discussion” sections of the manuscript, interpreted all results more cautiously, and highlighted the need for verification in larger future studies (Lines 738-747). The relatively small sample size was a result of balancing animal ethics (the 3R principles) and the practical challenges of the experimental complexity (attrition due to viral modeling and exercise interventions). We are confident that, under these constraints, the data obtained are robust enough to address the main scientific questions of this study.

The revised part: However, several limitations must be acknowledged. First, we did not examine whether miR-27a suppression improves inflammation or fibrosis in MS, which are critical to disease progression. Second, potential synergistic effects involving other miRNAs in the regulatory network cannot be ruled out. Third, although effect sizes were observed for primary outcomes, the sample size may limit power to detect smaller effects or complex interactions, increasing the risk of Type II errors. Consequently, caution is warranted when interpreting negative outcomes, and larger studies are needed to confirm these findings. Future research should use single-cell sequencing to clarify tissue-specific mechanisms and explore combined exercise-genetic strategies for improved clinical translation. (Lines 738-747)

• Uncut blots: The complete images for all western blots are provided in Supplementary information - Original gel blot images.

• Quantify miR-27a knockdown efficiency longitudinally: We appreciate the reviewer’s insightful comment regarding the longitudinal assessment of miR-27a knockdown efficiency. In the present study, due to technical limitations and animal welfare considerations (the 3R principle), we evaluated the knockdown efficacy at the experimental endpoint by measuring miR-27a expression in both skeletal muscle and liver tissues across all groups. While longitudinal measurements could provide additional kinetic information, our terminal assessment robustly confirmed significant miR-27a reduction in both silencing groups (SD-S-I and MS-S-I) compared with their respective null controls (SD-S-N and MS-S-N, P < 0.05), supporting the conclusion that effective and stable knockdown was achieved throughout the critical intervention period. Future studies employing longitudinal sampling or multiple time points would be valuable to further characterize the time-dependent dynamics of miR-27a suppression and its functional consequences.

2.Discussion expansion

• Human context and Mechanistic hypothesis: We thank the reviewer for this valuable suggestion. In response, we have expanded the Discussion section to include a comparison with human miRNA studies and have proposed a potential mechanism linking exercise to miR-27a downregulation. Specifically, we added clinical evidence showing elevated serum miR-27a levels in patients with metabolic syndrome and type 2 diabetes, and introduced exosome-mediated sorting and secretion as a plausible mechanism through which aerobic exercise may reduce miR-27a content (Lines 552-555, Lines 577-590).

3.Clarity improvements

We have simplified the abstract for enhanced readability (Lines 2-11) and standardized figure labels.

The revised part:

Abstract: This study investigated how aerobic exercise improves metabolic disorders in rats with metabolic syndrome (MS) and explored the role of miR-27a in this process. MS was induced by a high-fat and high-sugar diet. After eight weeks of treadmill training, aerobic exercise was found to reduce metabolic abnormalities and decrease elevated miR-27a levels, while increasing the expression of PPARγ and key downstream molecules involved in glycolipid metabolism. Downregulating miR-27a via a viral vector produced benefits similar to those of exercise, and combining miR-27a inhibition with exercise led to further improvement. These results suggested that aerobic exercise alleviates MS-related metabolic disorders partly through suppressing miR-27a and promoting PPARγ signaling, revealing a potential therapeutic target. (Lines 2-11)

4.Comparison to field & unresolved questions

Regarding this recommendation, we have added a section on strengths and limitations at the end of the discussion section (Lines 735-747).

The revised part: A key strength of this study is the translation of previous in vitro results to an in vivo setting, confirming PPARγ upregulation-beyond IRS-1 inhibition-as a central mechanism, as reported by Yu et al. [14], and validating the regulatory function of miR-27a under physiological conditions, consistent with Chen T et al. [34]. However, several limitations must be acknowledged. First, we did not examine whether miR-27a suppression improves inflammation or fibrosis in MS, which are critical to disease progression. Second, potential synergistic effects involving other miRNAs in the regulatory network cannot be ruled out. Third, although effect sizes were observed for primary outcomes, the sample size may limit power to detect smaller effects or complex interactions, increasing the risk of Type II errors. Consequently, caution is warranted when interpreting negative outcomes, and larger studies are needed to confirm these findings. Future research should use single-cell sequencing to clarify tissue-specific mechanisms and explore combined exercise-genetic strategies for improved clinical translation. (Lines 735-747)

Review Comments to the Author

Reviewer 1

---

## [Decision Letter · Decision Letter 1]

5 Jan 2026

The Mechanism of Aerobic Exercise in Ameliorating Glycolipid Metabolic Disorders in Metabolic Syndrome Rats via the miR-27a-PPARγ Pathway

PONE-D-25-28234R1

Dear Dr. Lin,

We’re pleased to inform you that your manuscript has been judged scientifically suitable for publication and will be formally accepted for publication once it meets all outstanding technical requirements.

Kind regards,

Rashmi Supriya

Academic Editor

PLOS One

Additional Editor Comments (optional):

Reviewers' comments:

Reviewer's Responses to Questions

**Comments to the Author**

Reviewer #2: All comments have been addressed

Reviewer #3: All comments have been addressed

2. Is the manuscript technically sound, and do the data support the conclusions?

Reviewer #2: Yes

Reviewer #3: Yes

3. Has the statistical analysis been performed appropriately and rigorously?

Reviewer #2: Yes

Reviewer #3: Yes

4. Have the authors made all data underlying the findings in their manuscript fully available?

Reviewer #2: Yes

Reviewer #3: Yes

5. Is the manuscript presented in an intelligible fashion and written in standard English?

Reviewer #2: Yes

Reviewer #3: Yes

Reviewer #2: The manuscript presents a compelling investigation into the miR-27a-PPARγ pathway as a key mechanism through which aerobic exercise ameliorates glycolipid metabolic disorders in a rat model of metabolic syndrome. The experimental design is robust, the methodology is sound, and the results are clearly presented and support the conclusions. This study provides valuable insights into the molecular basis of exercise-induced metabolic improvements and is a significant contribution to the field.

Reviewer #3: This study investigates how aerobic exercise improves metabolic disorders in rats with

metabolic syndrome (MS) and explores the role of miR-27a in this process. The authors have adequately addressed previous concerns. I have no other questions.

**Do you want your identity to be public for this peer review?** For information about this choice, including consent withdrawal, please see our Privacy Policy

Reviewer #2: No

Reviewer #3: **Yes:** Wei Yan

---

## [Editor Report · Acceptance letter]

PONE-D-25-28234R1

PLOS One

Dear Dr. Lin,

I'm pleased to inform you that your manuscript has been deemed suitable for publication in PLOS One. Congratulations! Your manuscript is now being handed over to our production team.

Kind regards,

on behalf of

Dr. Rashmi Supriya

Academic Editor

PLOS One